# Joint Beamforming Design for RIS-Assisted Integrated Satellite-HAP-Terrestrial Networks Using Deep Reinforcement Learning

**DOI:** 10.3390/s23063034

**Published:** 2023-03-11

**Authors:** Min Wu, Shibing Zhu, Changqing Li, Yudi Chen, Feng Zhou

**Affiliations:** 1School of Space Information, Space Engineering University, Beijing 101416, China; 1800022837@pku.edu.cn (M.W.); lcqqcl5577@sohu.com (C.L.); chenyudi9438@163.com (Y.C.); 2College of Information Engineering, Yancheng Institute of Technology, Yancheng 224051, China; zfycit@ycit.edu.cn

**Keywords:** reconfigurable intelligent surface (RIS), integrated satellite–HAP–terrestrial networks (IS-HAP-TNs), deep reinforcement learning (DRL), optimization performance

## Abstract

In this paper, we consider reconfigurable intelligent surface (RIS)-assisted integrated satellite high-altitude platform terrestrial networks (IS-HAP-TNs) that can improve network performance by exploiting the HAP stability and RIS reflection. Specifically, the reflector RIS is installed on the side of HAP to reflect signals from the multiple ground user equipment (UE) to the satellite. To aim at maximizing the system sum rate, we jointly optimize the transmit beamforming matrix at the ground UEs and RIS phase shift matrix. Due to the limitation of the unit modulus of the RIS reflective elements constraint, the combinatorial optimization problem is difficult to tackle effectively by traditional solving methods. Based on this, this paper studies the deep reinforcement learning (DRL) algorithm to achieve online decision making for this joint optimization problem. In addition, it is verified through simulation experiments that the proposed DRL algorithm outperforms the standard scheme in terms of system performance, execution time, and computing speed, making real-time decision making truly feasible.

## 1. Introduction

As fifth-generation mobile communication systems enter commercial operations worldwide, terrestrial wired and wireless networks are beginning to provide instant, high-speed data transmission services to users in high-density population areas. However, due to geographical conditions and business models, networks in remote areas are still unable to meet the needs of multiple users for full-area coverage and ubiquitous access [1,2]. Compared with traditional terrestrial wireless communication systems, the satellite–aerial–ground integrated network (SAGIN) emerged as a high-potential infrastructure for future wireless communication networks that can establish seamless coverage and massive connectivity for the explosive growth of terrestrial users [3]. In SAGIN, the high-altitude platform (HAP)-based relaying communications were expected to be the primary choice for aerial communication compared to UAV-based relaying, owing to their lower operating costs, longer residence time, deployment flexibility and the number of communication devices they can carry. Thus, the integration of HAPs for enabling unobstructed connectivity of the integrated satellite–terrestrial networks (ISTNs) attracted widespread attention from academia and industry [4]. Nevertheless, integrated satellite high-altitude platform terrestrial networks (IS-HAP-TNs) also raise serious concern about the rapidly growing energy consumption and wireless security in the transmission process, which are of great significance for maintaining green and reliable communication schemes.

Among the various candidates, a novel energy-efficient mode, known as reconfigurable intelligent surface (RIS), has widely been applied to improve communication security and network performance [5,6]. Each of the RIS reflective elements is a varactor diode that allows the amplitude and/or phase shift of the incident signal to be independently controlled by an embedded RIS central controller [7]. An extensive study in [8] showed that RIS has already been applied in many different communication network scenarios, such as ambient reflectors, signal transmitters, and even signal receivers. Meanwhile, RIS was also used in ambient forward scatter/backscatter communication systems, which is a seminal contribution, as in [9].

The ability of RIS to reconfigure transmission paths in real time with low cost provides a new solution to the time-varying radio channels caused by the high maneuverable IS-HAP-TNs and the severe path loss caused by long-distance propagation [10]. Furthermore, unlike traditional active beamforming in satellite communications, where additional active antennas or radio-frequency (RF) chains are installed, in the RIS-assisted IS-HAP-TNs, where only passive reflective elements are used for phase and amplitude adjustment and control, RIS enables full duplex passive beamforming with no hardware modifications and low energy costs [11]. The numerous quality features of RIS have prompted many researchers to carry out numerous applications in a variety of different communication system applications, such as an orthogonal frequency division multiplexing (OFDM) system [12], covert communication [13], multi-relaying networks [14], non-orthogonal multiple access (NOMA) [15] and so on. In recent years, several research efforts have gradually introduced the RIS concept into SAGINs, for example, by placing the RIS under the satellite’s solar panel, on the side of the UAV/HAP or close to the ground receiving users, in order to improve the system’s communication performance [16].

On the other hand, there are several advantages to installing the RIS on the side of HAP, rather than the traditional way of installing RIS on the ground buildings. Firstly, although HAPs are likely to establish LoS links in most cases, they still face the challenge of blockages caused by obstacles. In addition, the satellite-HAP-ground transmission link may be exposed to severe interference and malicious eavesdropping. To address these issues, RIS can be employed at HAP to strengthen the desirable signals at the intended users and overcome blockages by adjusting the phase shift. Moreover, RIS is able to deteriorate the unfavorable signals at the unintended users to mitigate the interference and information leakage. In [17], the authors considered RIS-assisted air–ground networks in two case studies. In the first case, the RIS was mounted on a UAV to improve the communication quality from the base station to the user. In the second transmission scenario, the RIS was used to assist the transmission from the UAV to the ground. Meanwhile, the authors in [18,19] considered all the auxiliary communication scenarios in which UAV carries RIS. From the aspect of RIS implementation, with the help of HAP, the RIS can be deployed more flexibly with an elevated position, where the LoS links between RIS and the transceiver ends can be easily established, especially in the crowded urban scenario. Secondly, since the transmission signal is more likely to block, movable RIS are better suited to enhancing communication than fixed RIS. A persistent LOS link to the transmitter and receiver is maintained by constantly adjusting the position of the reflector to changes in the environment.

Thus, assuming that RIS is installed on HAP, there will be three main modes of communication. One case is that HAP serves directly as a communication relay, the second is that RIS can be installed on the side of HAP for auxiliary communication, and the third is that both HAP and RIS serve as communication relays. Motivated by the above analysis, in our paper, the HAP only serves as a mobile platform and is equipped with RIS to increase line-of-sight transmission. This will lay a foundation for subsequent research.

However, the demanding requirements of emerging applications in RIS-assisted IS-HAP-TNs are difficult to meet using traditional mathematical optimization algorithms alone, including communication scenarios in satellite communications that are too complex due to unknown channel models and communication scenarios that cannot be described by accurate mathematical expressions, such as the inevitable non-linearities due to hardware losses [20,21]. Additionally, as for the dynamic RIS configuration scenario, the RIS reflection coefficients are assumed to be reconfigured many times during one transmission instance. The capacity region that corresponds to the dynamic RIS configuration is obtained by the union of all achievable rate tuples over all possible combinations of values of reflection coefficients for each RIS element. However, determining this capacity region is prohibitively complex since the size of the set of all possible combinations of reflection coefficients grows exponentially with the number of reconfiguration. Considering that the future IS-HAP-TNs needs to meet a large number of real-time Internet of Things (IoT) or Internet of Vehicles (IoV) but, due to its channel environment, is complex and highly dynamic, the calculation optimization time of conventional methods is long, so more intelligent algorithms are urgently needed.

On the contrary, deep reinforcement learning (DRL) is a novel approach that combines deep learning (DL) and reinforcement learning (RL). It has been proven to be a significant breakthrough in non-convex optimization problems, including hybrid beamforming design [22], spectrum intelligence sensing [23], channel state estimation [24], and power allocation strategy optimization [25]. Compared with deep learning (DL), the DRL algorithm does not require a large amount of training labeled data as inputs and is therefore very friendly for the optimization of wireless communication systems, where obtaining data is more tedious. By interacting with the environment to obtain rewards from the network, DRL can learn and construct wireless channel knowledge without knowing the complete channel model information and the precise movement pattern, while implementing efficient algorithm design through embedded neural networks to sequentially find optimal solutions to complex multi-objective optimization problems. In [26], a deep Q-network (DQN) with greedy characteristics is proposed for the joint optimization of beamforming design, power allocation strategy and interference coordination for maximizing the signal to interference plus noise ratio (SINR). In [27], by using the DRL framework, the user distribution model is tracked and predicted to autonomously and dynamically optimize the MIMO broadcast beam and propose the optimal broadcast beam for each served cell. The results confirm that optimal coverage can be achieved using the DRL framework in both single-sector and multi-sector environments, and in both periodic and Markovian mobility modes.

### 1.1. Related Work and Motivation

Nowadays, many prior works are developed to investigate the performance of RIS-assisted wireless communication.
(1)For the RIS-assisted wireless communication: In the future wireless communications, the RIS can be deployed as the reconfigurable signal transmitter, receiver and passive reflector array. Thus, by optimizing the RIS reflector element phase shifts, for example, by selecting optimal reflector element positions and passive beamforming designs, RIS-assisted intelligent radio environments are a promising optimization paradigm to transform the design of modern wireless networks. In [28], the author jointly optimized the RIS reflection coefficients and the number of tunable reflective elements to improve the quality of service (QoS) of RIS-assisted edge networks by considering the actual amplitude and phase shift model of the RIS. The authors in [29] proposed an efficient alternating algorithm for fractional programming (FP), majorization–minimization (MM), and manifold optimization methods to jointly optimize active and passive beamforming under multiple constraints of radar sensing similarity, RIS constraints and transmit power constraints. Moreover, the authors in [30] considered the performance of RIS-assisted integrated satellite unmanned aerial vehicle (UAV)-terrestrial networks, and the closed-form expression for the outage probability (OP) was obtained to evaluate the impact of the introduction of RIS on the system performance.(2)For the DRL in RIS-assisted wireless communication: By combining the function fitting benefits of deep learning with the environmental interaction decision-making benefits of reinforcement learning, DRL is believed to have the ability to solve non-linear problems without the need for a prior relaxation, enabling direct solutions to non-linear problems in mathematical methods. Recently, DRL-based methods were also applied to solve RIS-assisted wireless communication optimization problems. The authors in [31] presented a RIS-assisted multi-user full-duplex secure communication system with hardware impairments and maximized the total secrecy rate by using DRL. In [32], the authors considered a deep deterministic policy gradient (DDPG)-based framework to tackle the problem of maximizing the receiver signal-to-noise ratio (SNR) in a RIS-assisted single-user wireless communication system. The simulation results showed that the proposed DDPG algorithm can achieve higher performance in a shorter running time compared to traditional methods based on semi-definite relaxation.

Based on the advantages of DRL itself, the model-free DRL emerged as an extraordinarily remarkable technology to address massive data, mathematically intractable non-linear non-convex problems and high-computation issues. The DRL technology is most appealing to large-scale MIMO systems, such as the satellite communication or the IS-HAP-TNs with a massive number of array elements, where optimization problems become non-trivial due to the extremely large dimension optimization involved. Actually, DL-based approaches are able to significantly reduce the complexity and computation time utilizing the offline prediction but often require an exhaustive sample library for online training. Meanwhile, the deep reinforcement learning (DRL) technique, which embraces the advantage of DL in neural network training as well as improving the learning speed and the performance of reinforcement learning (RL) algorithms, has also been adopted in designing wireless communication systems. All of the above work verifies that the DRL algorithm can effectively solve the optimization problems of various wireless communication systems in terms of different performance metrics and achieve high performance in a short run-time with an appropriate algorithmic framework design.

### 1.2. Major Contributions and Novelty

Currently, the joint beamforming design technique is also a hot issue, which can greatly improve the communication efficiency, system capacity and transmission rate of wireless communication systems to some extent. The efficiency of applying DRL to RIS-assisted wireless communication systems has been demonstrated in simulation experiments in the relevant literature. Related meta-surface techniques, including RIS, have also recently been widely used for IS-HAP-TNs [33,34,35,36].

Motivated by the above analysis, we investigated the joint beamforming design for a RIS-assisted IS-HAP-TNs, in which more constraints must be considered, such as the angle of arrival (AoA) or the angle of departure (AoD) in RIS, maximum transmit power of the satellite, the optimal RIS phase shift matrix to reflect, and the highly dynamic signal transmission environment. However, the statistical characteristics of the shadowed-Rician (SR) channel are far more complex than the terrestrial Rayleigh channel, which is also the difficulty and innovation of our work. Nevertheless, since the IS-HAP-TNs environment is harsh as well as time varying, and the hardware of RIS may suffer from the damage unpredictably, it is challenging to perceive accurate and complete channel state information (CSI), making conventional optimization-based methods to IS-HAP-TNs operation no longer appropriate. In addition, the conventional optimization-based methods need a rich body of iterations to achieve a satisfactory solution, which results in it being impracticable for making real-time decisions in time-varying IS-HAP-TNs. Targeted at solving the difficult problem under uncertainty, deep reinforcement learning (DRL) emerged as the times required and was applied to solve the problems in RIS-aided wireless communications. By applying the advanced DRL framework to optimize the ground user transmit beamforming and RIS phase shift matrix to send the desired signal to the target satellite, the overall network system sum rate is maximized [37]. To the best of the authors’ knowledge, there is no prior work focusing on the beamforming design for RIS-assisted integrated satellite-HAP-terrestrial networks, especially the installation of RIS on HAP, which motivates our work. In this paper, considering the high dynamics of RIS-assisted IS-HAP-TNs transmission process, the main work and contributions are summarized as follows:(1)Firstly, an innovative system model of installing RIS on the HAP side is proposed in the IS-HAP-TNs. Considering the time-varying characteristics of the IS-HAP-TNs fading channel model and signal transmission model, the system sum rate formulations are given under these system model constraints using the active transmit beamforming at the ground user equipment, the phase shift matrix at the RIS, and the maximization expressions under the proposed constraints.(2)Secondly, a parameter soft-updated strategy framework based on DDPG framework is designed to optimize the above system sum rate maximization problems. The framework does not need to know the explicit model and specific mobile model of the wireless environment, and solves the formal problem of the system by rationalizing the design of the elemental state space, action space and reward function in the DDPG algorithm so that it can handle continuous variable problems well.(3)Finally, the simulation experiments on the number of RIS elements as well as the average reward show that the designed DRL algorithm framework outperforms other comparative baseline algorithms, which illustrates the effectiveness of the DRL algorithm in solving joint beam optimization problems and provides guidance for real-time decision making in dynamic IS-HAP-TNs communication environments.

The remainder of this paper is arranged as follows. Section 2 describes the considered system model and identifies the optimization objective problem under the constraints. Section 3 gives the basic framework of the soft update parameter strategy and gives the design flow for the optimization of the active transmit beamforming matrix and the RIS phase shift matrix under this framework. Section 4 plots the network performance simulation results under this framework and provides a detailed theoretical analysis. Finally, Section 5 concludes the whole work.

## 2. System Model Description

In this illustration, we envision an uplink transmission communication system that includes a geosynchronous Earth orbit (GEO) satellite and backward high altitude platforms (HAPs) deployed with RIS, as well as the *K* ground user equipment (UE) employing a single antenna as shown in Figure 1. In our proposed system model, the UEs transmission communication information through RF links to the RIS with *M* reflective elements installed on the HAP, which acts as a passive reflection relay with changeable transmission links and sends the received signal to the satellite.

It is noted that the satellites are linked to the cloud data computing processing center by the free-space optical (FSO), which can collect global communication information, such as the user’s requirements, as the system control link. Instead of coding satellites and HAPs separately, it centralizes the baseband processing of the entire network in the cloud, with the cloud as the core, taking into account resource management and environmental feedback [38].

In order to realistically simulate the UEs-RIS link, where the RIS is mounted on the HAP in the aerial, here, we consider the small-scale path loss model [39], and then, the channel model vector of the UEs-RIS can be expressed as
(1)$$h_{UR}=\sqrt{\frac{MK}{L_{total}}}\sum_{l=1}^{L_{total}}\alpha_{l}g\left(m,\varphi_{AR}\right)g^{T}\left(k,\varphi_{DU}\right)$$
where $L_{total}$ denotes the number of the total transmission path, $\alpha_{l}$ represents the Nakagami-*m* channel model random variable, $\varphi_{AR,l}$ and $\varphi_{DU,l}$ denote the angle of arrival (AoA) of RIS and the angle of departure (AoD) of the UEs in the *l*-th transmission path. The channel model vector $g\left(L,\varphi \right)$ as a function of the transmission path *L* and the AOA or AOD φ can be expressed as
(2)$$g\left(L,\varphi \right)≜\frac{1}{\sqrt{L}}{\left[1,e^{j\pi \cos\varphi },e^{2j\pi \cos\varphi },...e^{\left(L-1\right)j\pi \cos\varphi }\right]}^{T}$$

The RIS-satellite uplink channel vector is denoted by $H_{RS}$, which can be expressed as
(3)$$H_{RS}=\sqrt{MN_{s}P_{r}}\left[g\left(N_{s},\varphi_{AS}\right)\right]g\left(M^{T},\varphi_{DR}\right)$$
where the $N_{s}$ denotes the antenna numbers of the uniform linear array (ULA) in the satellite, and $\varphi_{AS}$ and $\varphi_{DR}$ are the AOA of the satellite and the AOD of the RIS, respectively. Meanwhile, the $P_{r}$ is the free space path loss between the RIS and the satellite [40]. Note that in the RIS-satellite uplink channel model, considering that the HAP flies at a higher altitude than most ground buildings and the RIS is mounted on the HAP, we only assume the line-of-sight (LoS) transmission path between the RIS and the satellite, and the $P_{r}$ can be expressed by the following formula:(4)$$\begin{array}{c} P_{r}=\frac{\lambda^{2}G_{sr}G_{st}}{{\left(4\pi \right)}^{2}{d_{sr}}^{2}\kappa_{a}T_{a}B_{W}} \end{array}$$
where λ, $G_{sr}$, $G_{st}$, $d_{sr}$, $\kappa_{a}$, $T_{a}$, $B_{W}$ denote the carrier wavelength of signal, the gains of every RIS reflection unit, antenna gain of each satellite, the transmission distance between RIS to center of satellite coverage area, the Boltzmann constant, the temperature of the propagating noise and the frequency band of signal, respectively. The direct uplink channel $H_{US}$ from the UE and the satellite is basically a standard MIMO channel model and can be characterized by existing methods to express its channel characteristics [41].

### 2.1. Signal Transmission Model

We assume that the *k*-th UEs intends to transmit signals denoted as $\sqrt{w_{k}}s_{k}$, where $w_{k}$ is the transmission power matrix coefficient vector at the *k*-th UEs of the total transmit beamforming matrix $W=[w_{1},w_{2},...,w_{k}]$ under less than the maximum power constraint $P_{max}$ and $s_{k}$ is satisfied $E\left[\left|s_{k}{\left(t\right)}^{2}\right|\right]=1$ with zero mean unit variance entries at the *t*-th transmission moment [42]. The transmitted signal from the ground user propagates through the direct link and the reflected link, the latter link reaching the satellite under the reflection of the RIS mounted on the HAP by means of changing the phase of the RIS reflecting elements. We define that Φ denotes the phase shift diagonal matrix applied to the reflective RIS for input by $\Phi \left(m,m\right)=\phi_{m}=\chi_{m}e^{j\varphi_{m}}$, where $\chi_{m}$ is the magnitude and $\varphi_{m}$ denotes the phase shift caused by the passive reflection of each reflective element of the RIS [43]. Thus, the signal received by the satellite at the *t*-th time step can then be further expressed in the following:(5)$$\begin{array}{c} y\left(t\right)=\left(H_{US}^{k}+H_{RS}^{k}\Phi h_{UR}^{k}\right)\sqrt{w_{k}}s_{k}+\sum_{j,j\neq k}^{K}\left(H_{US}^{j}+H_{RS}^{j}\Phi h_{UR}^{j}\right)\sqrt{w_{j}}s_{j}+n_{0} \end{array}$$
where $n_{0}$ denotes the system zero mean additive white Gaussian noise (AWGN) followed by $n_{0}∼CN\left(0,\sigma_{n}^{2}\right)$. As we can see from Equation (Equation 5) above, the introduction of the RIS does not introduce additional AWGN compared to the traditional use of relay-based communication systems. This is because the RIS is only a passive mirror relay that only reflects the signal incident on its plane without signal decoding and encoding. Once the RIS receives the signal, its phase will reconfigure the signal by means of a central microprocessor controller connected to the RIS. Thus, the received signal-to-interference-plus-noise ratio (SINR) of $UE_{k}$ signal at the satellite is given by
(6)$$\begin{array}{c} \gamma_{k}=\frac{{\left|\left(H_{US}^{k}+H_{RS}^{k}\Phi h_{UR}^{k}\right)\right|}^{2}w_{k}}{\sum_{j,j\neq k}^{K}{\left|\left(H_{US}^{j}+H_{RS}^{j}\Phi h_{UR}^{j}\right)\right|}^{2}w_{j}+\sigma_{n}^{2}} \end{array}$$

The system can be described that the UEs transmits signals to satellite, so we set the performance metric for evaluating RIS-assisted wireless systems as the system ergodic sum rate, which can be modeled as
(7)$$\begin{array}{c} C\left(H_{US},H_{RS},\Phi ,w,h_{UR}\right)=\underset{k=1}{\sum^{K}}\;R_{k}=\log\left(1+\frac{\sum_{k=1}^{K}{\left|\left(H_{US}^{k}+H_{RS}^{k}\Phi h_{UR}^{k}\right)\right|}^{2}w_{k}}{\sigma_{n}^{2}}\right) \end{array}$$
where $R_{k}$ represents the data transmission rate of the *k*-th UEs, formulated by $R_{k}=\log_{2}\left(1+\gamma_{k}\right)$.

### 2.2. Problem Formulation

In this section, the aim is to determine the optimization objective function as the maximum rate of chemistry for the RIS-assisted IS-HAP-TNs involved. From the above, it is clear that $W=[w_{1},w_{2},...,w_{k}]$ denotes the transmit beamforming matrix vector, and the proposed optimization problem can be modeled as
(8)$$\begin{array}{cc} \max_{\left\{W,\Phi \right\}}\; & C\left(H_{US},H_{RS},\Phi ,w,h_{UR}\right)\\ s.t. & \;C1:\;w_{k}\leq P_{\max},\forall k=1,2,...,K,\\ \; & \;C2:\;\left|\phi_{m}\right|=1,\forall m=1,2,...M. \end{array}$$
where $P_{max}$ represents the maximum link transmission power. The constraint C1 regulates the transmission maximum power of UEs. The constraint C2 represents the constraints on RIS reflective elements. Obviously, the above optimization problem based on the proposed system model is non-convex, while constraints exist for a non-convex non-trivial optimization problem owing to the high-dimensional reflection elements phase shift and can hardly be settled by traditional improvement approaches. If the traditional mathematical tools were used, the problem would have to be solved by an exhaustive search to obtain the optimal solution, which at the same time implies a large amount of computational resources and processing measures, which is almost impossible for large-scale network communication scenarios, especially for networks with high real-time requirements such as our considered IS-HAP-TNs. Thus, this paper only considers the use of intelligent algorithms for the optimal beamforming solution, it does not mathematically solve challenging optimization problems by non-convex to convex conversions, etc.

Meanwhile, to demonstrate the superiority of the proposed algorithm, we refer to the traditional alternating optimization (AO) algorithm in the literature for solving the joint active transmit beamforming and passive beamforming optimization algorithm. First, the constraint in Equation (Equation 9) can greatly increase the difficulty of solving this problem, so it is also necessary to relax this constraint to $\left|\phi_{m}\right|\leq 1,\forall m=1,2,...,M$. After obtaining the local optimal solution, the reflection coefficient is changed back to a value that conforms to the modulus constraint of 1 by the projection method. In each iteration of traditional AO algorithms, the globally sub-optimal W is solved by first fixing Φ and the sub-optimal Φ is solved by fixing the matrix W until the algorithm converges. For the design of high-dimensional continuous variables, such as the each element phase shift in large-scale networks, including the transmit power matrix, the phase shift matrix, etc., traditional mathematical optimization methods such as the AO algorithm and water filling (WF) algorithm cannot effectively solve these problems and often generate local optimal deviations. Thus, to efficiently tackle the considered non-convex jointly optimization problems, we propose a soft-updated DDPG algorithm, where, through continuous trial-and-error interaction with the environment, the DRL agent gradually learns a deterministic strategy that leads to the optimal action.

## 3. Soft-DDPG-Based Joint Active and Passive Beamforming Design

In this section, the method of DRL is used to jointly optimize the transmit beamforming shape and phase shift array, and utilizing the DDPG structure shown in Figure 2. First, we briefly discuss the soft-DDPG principle and operation process. Then, we introduce the proposed DRL architecture and provide a detailed description of the state, action, reward, and the algorithm framework.

### 3.1. Overview of Soft-DDPG

It is supposed that there exists a central controller or a learning agent in this network that can collect the channel information or communication date immediately, such as the RIS to satellite channel $H_{RS}$ and $h_{UR}$ and the UE to the RIS channel $h_{UR}$. Figure 2 displays the soft-DDPG architecture suitable for the earning agents to interact with high dynamic communication environments to obtain the pre-defined rewards or punishments. The core concept of the soft-DDPG framework proposed in this letter is to perform effective beamforming design and phase shift convert under unforeseen circumstances, such as local state observations, e.g., RIS. The algorithm mainly includes two kinds of deep neural networks (DNNs), namely, the training network and the target network. To avoid or mitigate the issue of updating state participant values in a single case, we assume that the target and training networks have the same neural network architecture.

Based on the above extensions, we can more clearly portray the framework covered in this article, with four DNNs drawn in detail, which are the training critic network, the training actor network, the target critic network and the target actor network. The functions of these four neural networks described above are described below. The training critic network needs to input the current state $s^{\left(t\right)}$ into the action network and output the current action $a^{\left(t\right)}$, and the training actor network needs to input state $s^{\left(t\right)}$ and action $a^{\left(t\right)}$ into the training critic network and output the Q value $Q_{\pi }\left(s^{\left(t\right)},a^{\left(t\right)}\right)$. The target critic network needs to input the updated state $s^{\left(t+1\right)}$ to the target actor network and output the $a^{\left(t+1\right)}$. The target actor network needs to input the updated $s^{\left(t+1\right)}$ and $a^{\left(t+1\right)}$ to the target critic network and output the target Q value $Q_{\pi }\left(s^{\left(t+1\right)},a^{\left(t+1\right)}\right)$.

Considering the existence of plural inputs in the neural network input, this proposed model uses tanh as the activation function of the hidden layer to limit the action space in the interval (0,2π), and to eliminate the effect of the change in the distribution of the hidden layer data brought by the parameter update. This proposed DRL framework introduces a batch normalization layer after each hidden layer to process its output. The batch normalization layer can effectively combat the gradient disappearance phenomenon, improve the training efficiency, and make the training process of the deep layer network more stable. In addition, according to the constraints of the transmitting power and phase shift coefficients, the proposed model adds the tanh activation function to the output layer of the actor network to restrict the output to the interval [−1,1], and subsequently transforms the action into the data format required by the optimization problem by taking the absolute value normalization and range mapping methods to meet the constraints of the power allocation and phase shift so as to calculate the system sum rate as Equation (Equation 8).

We generate different transmission link channel information by following the channel model features described earlier when channel state information (CSI) and the previous action $W^{\left(t-1\right)}$ and $\Phi^{\left(t-1\right)}$ are known at the *t*-th time step, and the learning agent can establish knowledge about the current state space $s^{\left(t\right)}$ in the *t*-th time step. It is considered that the difficulty of the joint optimal design of active transmission beamforming and passive RIS phase shift matrix are discrete and present a great challenge to continuous state-space and action-space settings. Next, the details of the DRL-based algorithm state space *S*, action space *A* and instant reward function *R* are explained below.

State: State space is generally a description of the environmental observations at the *t*-th time step. In this paper, the DRL algorithm state space includes three parts, i.e., the *k*-th UE’s transmission power at the (t−1)-th time step, the CSI of all communication links including direct links and cascaded reflective links containing RIS reflective elements at the (t−1)-th time step, and the action from the (t−1)-th time step, which are represented by $w_{k}^{\left(t-1\right)}$, G and $a^{\left(t-1\right)}$, respectively, where
(9)$$\begin{array}{c} G≜\left[H_{US},H_{RS},H_{{}_{RS}}^{H},h_{UR},h_{UR}^{H}\right] \end{array}$$

Thus, the state space can be expressed as $s^{\left(t\right)}=\left[w_{k}^{\left(t-1\right)},G,a^{\left(t-1\right)}\right]$.

Action: Action space is generally a series of choices for the next action. Once the agent performs the current action $a^{\left(t\right)}$ step by step during the learning process according to the transfer policy π at the *t*-th time slot, the state space of the environment will be shifted from $s^{\left(t\right)}$ to the next state $s^{\left(t+1\right)}$. The action space is designed as the UE transmit power matrix W and the RIS phase shift matrix Φ. Considering that neural networks can only take real part as input and and match the neural network input formats, the process of constructing the action space, where both the transmit power imaginary distribution, such as W=ReW+ImW and Φ=ReΦ+ImΦ are to be separated as separate input ports. Thus, the action space can be expressed as
(10)at=Rewkt,Imwkt,Reϕmt,Imϕmt
where the $w_{k}^{\left(t\right)}$ represents the *k*-th UE transmit power matrix, and the $\phi_{m}^{\left(t\right)}$ denotes *m*-th reflective element’s phase shift at the *t*-th time step.

Reward: The purpose of this paper is to maximize the system sum rate, and Equation (Equation 10) is adopted as the reward function:(11)$$\begin{array}{c} r^{\left(t\right)}=\;C\left(H_{US},H_{RS},\Phi ,w,h_{UR}\right) \end{array}$$

### 3.2. The Process of Algorithm Training

In order to break the coupling between experiences and adapt to a high dynamic environment, the experience replay approach allows agent access to previous historical experiences in subsequent training, the DDPG framework considered in this article. For policy-based algorithms, the agent collects experience in the episode. After an episode is run, experience is lost. It is better with a multi-threaded parallel architecture. This not only solves the previous problems but also makes efficient use of computing resources and improves the training efficiency.

In the proposed DDPG framework, the entire agent consists of a global network and multiple parallel independent workers, each including a set of an actor network and critic network. Each worker interacts independently with their own environment, gaining independent sampling experiences that are independent of each other, thus breaking the coupling between experiences to match the experience replay. Most of the underlying algorithms in DRL are single threaded, that is, a learning agent that interacts with the environment to generate experience. Including the underlying version of the actor network and critic network, because the environment is fixed and the action of the agent needs to be continuous, the experience gathered has strong timing associations and only part of the state and action space can be explored in a limited amount of time. To solve this problem, we adopt the soft-DDPG-based scheme to optimize the design process and present the corresponding pseudo-code in Algorithm 1.

In the initial stage of the algorithm, the experience replay buffer *D*, the training actor network $\psi \left(\cdot \right)$ and training critic network $Q\left(\cdot \right)$ need to be initialized randomly (Lines 1–2). They are copied to the target network $\psi^{\prime }\left(\cdot \right)$ and $Q^{\prime }\left(\cdot \right)$ (Line 3). After initializing and randomly generating the RIS-assisted IS-HAP-TNs communication channel environment state, the state is processed via DNN and the output $Y_{t}$ (Lines 5–6). The action is derived based on $Y_{t}$, where N is denoted as random noise, with the aim of seeking efficient exploration (Lines 7–8). In this letter, we employ the mini batch to reduce the sample training amount of sampling and ensure the quality of gradient reduction. After the transformation sequence is saved in the memory replay buffer *D* (Line 10), to achieve the optimal action that maximizes the output of the critic train network, the two train networks are updated using the minibatches of size ζ randomly sampled from replay buffer *D* (Line 11). We update the critic target network parameters $Q\left(\cdot \right)$ by minimizing the variance loss (Line 14). We make use of linking rules to update the actor networks parameters $\psi \left(\cdot \right)$. Finally, the target networks parameters of the actor network and critic network are slowly soft updated using the control factor τ as the decaying rate (Line 15).
**Algorithm 1:** Soft-DDPG-based Algorithm1: Initialize experience memory *D* to empty;2: Randomly initialization generate actor target/training     network $\psi^{\prime }\left(\cdot \right)$ and critic target/training network $Q^{\prime }\left(\cdot \right)$
    with parameters $\xi_{a}^{\prime }$ and $\xi_{c}^{\prime }$, separately;3: **Input**: w, ϕ, $H_{RS}$ and $h_{UR}$;4: **Output**: Optimal action $a_{opt}^{\left(t\right)}$;5: **for** each episode **do**:6:  Initialize state $s^{\left(0\right)}\in S,S\leftarrow s^{\left(0\right)}$;7:  **for** t=0,1,2,...T−1 **do**:8:     Choose action $a^{\left(t\right)}=\pi \left(s^{\left(t\right)}|\theta^{\pi }\right)+N$;9:     Take action $a^{\left(t\right)}$, get reward $r^{\left(t\right)}$ and $s^{\left(t\right)}$ evolves into
new state $s^{\left(t+1\right)}$;10:   Save $\left(s^{\left(t\right)},a^{\left(t\right)},r^{\left(t\right)},s^{\left(t+1\right)}\right)$ into *D*;11:   Randomly sample ζ transitions form *D*;12:   Training framework via DNN;13:   Compute target value for the critic’s evaluation
network by    
$y^{\left(i\right)}=r^{\left(i\right)}+\gamma Q_{\pi^{\prime }}^{\prime }\left(s^{\left(i+1\right)},\pi^{\prime }\left(s^{\left(i+1\right)}|\theta^{\pi^{\prime }}\right)|\theta^{Q^{\prime }}\right)$
14:   Update the parameters of the critic’s evaluation
network by     $L\left(\theta^{Q}\right)=\frac{1}{\zeta }\sum_{i=1}^{\zeta }\;{\left(y^{\left(i\right)}-Q_{\pi }\left(s^{\left(i\right)},a^{\left(i\right)}|\theta^{Q}\right)\right)}^{2}$;15:   Update the parameters of actor network with sampled policy gradients by    
$\nabla_{\theta^{\pi }}J=$
     $\frac{1}{\zeta }\sum_{i=1}^{\zeta }{⛛}_{a}Q_{\pi }(s,a|\theta^{Q}){|}_{a=\pi (s^{\left(i\right)}|\theta^{\pi })}{⛛}_{\theta^{\pi }\pi (s|\theta^{\pi })};$16:   Soft-update the parameters of DDPG’s target networks by    
$\theta_{c}^{\left(target\;\;\;\;\;\;\right)}\leftarrow \tau_{c}\theta_{c}^{\left(train\;\;\;\;\;\;\right)}+\left(1-\tau_{c}\right)\theta_{c}^{\left(target\;\;\;\;\;\;\right)}$
     $\theta_{a}^{\left(target\;\;\;\;\;\;\right)}\leftarrow \tau_{a}\theta_{a}^{\left(train\;\;\;\;\;\;\right)}+\left(1-\tau_{a}\right)\theta_{a}^{\left(target\;\;\;\;\;\;\right)};$17:   Update the state $s^{\left(t+1\right)}$;16:  **end for**;17: **end for**;

During each iteration *t* each of the learning process, the actor train network will select the action from the continuous action space based on the current state $s^{\left(t\right)}$. During this training process, in order to effectively explore the optimal action, the stochastic noise $N_{a}$ is also taken into account in the algorithm framework to obtain the deterministic strategy, i.e., $a^{\left(t\right)}=\pi \left(s^{\left(t\right)}|\theta^{\pi }\right)+N_{a}$, where $\theta^{\pi }$ is the actor train network parameter, and π is the transfer policy. When the operation ends, the environment will transit the last action to the next state $s^{\left(t+1\right)}$ to obtain instant reward $r^{\left(t\right)}$, and then obtain an evaluation for the action to evaluate the optimal action $a^{\left(t\right)}$, modeling a state–action value function by parameterized by $\theta^{Q}$ as
(12)$$\begin{array}{cc} \; & Q_{\pi }\left(s^{\left(t\right)},a^{\left(t\right)}|\theta^{Q}\right)\leftarrow \alpha Q_{\pi }\left(s^{\left(t\right)},a^{\left(t\right)}|\theta^{Q}\right)\\ \; & +\left(1-\alpha \right)\left[r^{\left(t\right)}+{\left.\gamma \max_{a^{\prime }}\;Q_{\pi }\left(s^{\left(t+1\right)},a^{\prime }|\theta^{Q}\right)\right|}_{a^{\prime }=\pi \left(s^{\left(t+1\right)}|\theta^{\pi }\right)}\right] \end{array}$$
where α denotes the algorithm learning rate in this algorithm framework. To ensure the stability, the target actor network is parameterized by $\theta^{\pi^{\prime }}$ and the target critic network is characterized by $\theta^{Q^{\prime }}$, which is parameterized at intervals according to the online network parameters. Thus, considering that the parameter update strategy is a soft update method, the algorithm is called soft-DDPG. The soft update method of parameters ensures the slow update of parameters and alleviates the instability problem of the policy network during the learning process.

## 4. Numerical Simulation Results

In this section, we evaluate the performance improvement of the proposed DRL-based algorithm framework for the proposed system model from different perspectives. First, we randomly generate a channel model matrix following the shadowed-Rician fading distribution and the corresponding channel models mentioned in the article [44]. The system parameters and hyperparameters of the DDPG algorithms are listed in Table 1 [45]. To test whether our algorithm improves network performance, we also consider three other standard solutions:(1)Hard-DDPG: The scheme indicates that the parameters in the DRL framework are updated in a hard-update strategy, which allows the network to copy all the parameters in the network at this time directly into the target network after every $t_{u}$ training session by pre-setting the parameter update interval $t_{u}$.(2)Random RIS: The scheme denotes that the RIS phase shift matrix Φ is randomly generated.(3)Without RIS: This scheme denotes that the communication scenario without RIS and the UEs can send signals directly to the satellite. Considering that the process is a continuous transmission, we assume that the successful transmission signal is 1/2.(4)Traditional optimization scheme: The traditional algorithm is an alternating optimization algorithm, the specific process is to obtain the transmit power matrix by fixing the RIS reflection coefficient, according to the water filling algorithm. Then we obtain the corresponding current jointly beamforming design according to Equation (Equation 9) until the objective function value converges.

Figure 3 plots the relationship between the number of RIS reflection elements and the system sum rate. As can be seen from the figure, the system sum rate is significantly higher for all algorithms as the number of RIS elements increases due to the fact that more RIS reflection elements increase the reflection channel gain but also sacrifice the complexity of the RIS deployment at the HAP. In addition, we can observe that the soft-update parameter strategy obtains a higher system sum rate than the hard update parameter strategy, alleviates the instability of the Q-value network in the learning process, and the soft-update strategy obtains a higher system ensemble rate by more flexibly interacting with the environment to design the phase shift matrix more flexibly. It can also be seen that the traditional alternating optimization algorithm used can also obtain a high system sum rate, and that the value of the objective function it obtains does not decrease as the number of elements increases, which also ensures the convergence of the algorithm.

The setting of hyper-parameters will have a great impact on the performance, such as the stability and convergence speed of neural networks. This paper also explores the effect of different learning rates on the performance and convergence speed of the model in our proposed DRL framework. The average reward is used to measure its performance, which can be shown as
(13)$$\;\;\;\;\;\;\;\mathrm{average}\;\;\;\_\;\;\;\mathrm{reward}\;\;\;\;\;\;\;\left(T_{i}\right)=\frac{\sum_{t=1}^{T}\;\;\;\;\;\;\;\;r^{\left(t\right)}}{T_{i}},T_{i}=1,2,\ldots ,T$$
where *T* is the maximum step size of sample training. All the parameters initialized in the training phase and the channel sample parameters used for different parameters are the same for this simulated neural network, and a comparison of the network performance at different learning rates, i.e., 0.01, 0.001, 0.0001, 0.00001 is shown in Figure 4. Figure 4 shows the average reward versus time step under different learning rates, and it can be seen that the effect of different learning rate settings in the neural network on the performance of the DRL algorithm varies greatly. For larger learning rates, such as 0.01, the performance of this network decreases, and the convergence effect is unstable and prone to oscillations. In particular, the considered DRL framework with a learning rate of 0.001 performs the best, but converges more slowly than the others. As the RIS reflection element increases, the average system reward also increases gradually as expected with the addition of reflection channels, but this does not significantly increase the convergence time of the proposed DRL framework [46]. In summary, DRL-based algorithms, such as the DDPG used in this paper, are very sensitive to the setting of hyper-parameters in the neural network, and the optimal hyper-parameter settings can vary considerably under different model structures [47]. Therefore, careful experimental tuning is required to obtain optimal parameter settings that can significantly improve the performance and convergence speed of the algorithm [48,49].

Figure 5 shows a schematic comparison of the average reward performance and the outdated CSI coefficients, respectively. In the proposed DRL framework, we choose the last moment CSI as the state-space input, and we can see that the average reward of all algorithms decreases gradually as the outdated CSI coefficient decreases. However, the proposed soft-DDPG framework remains at a favorable level compared to the existing scheme and the hard-DDPG scheme. Compared with the advanced DRL schemes, which do not require an exact channel model information, the existing alternating parameter optimization scheme relies on the knowledge of static exact channel model, but because of the high dynamic communication scenario, the system performance is not as good as the soft DDPG and hard DDPG schemes.

## 5. Conclusions

This paper discussed the joint optimal design scheme of transmitting active beamforming and passive beamforming for maximizing the system sum rate. In the IS-HAP-TNs assisted by RIS, it is hard to sense the channel state information in the dynamic environment accurately and comprehensively. On this basis, a novel type of DRL architecture is proposed, namely the soft-DDPG algorithm. With the help of the network parameter soft-update strategy, the coordination of the phase shift matrix can be obtained, even when the increasing number of RIS reflective element amplitudes changes. Simulation results show that the proposed framework can achieve better network performance in a lower operation duration and can be applied to the real-time control of IS-HAP-TN systems.

## Figures and Tables

**Figure 1 sensors-23-03034-f001:**
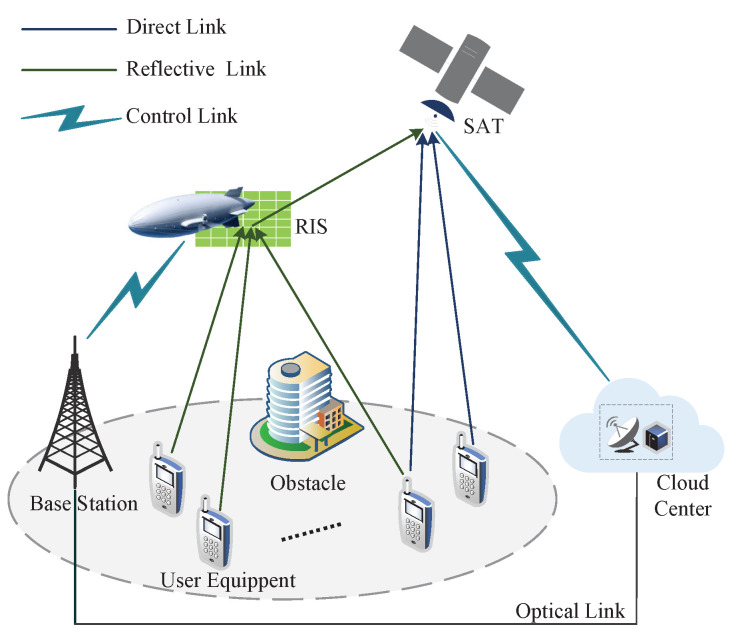
Illustration of a RIS-assisted IS-HAP-TNs system.

**Figure 2 sensors-23-03034-f002:**
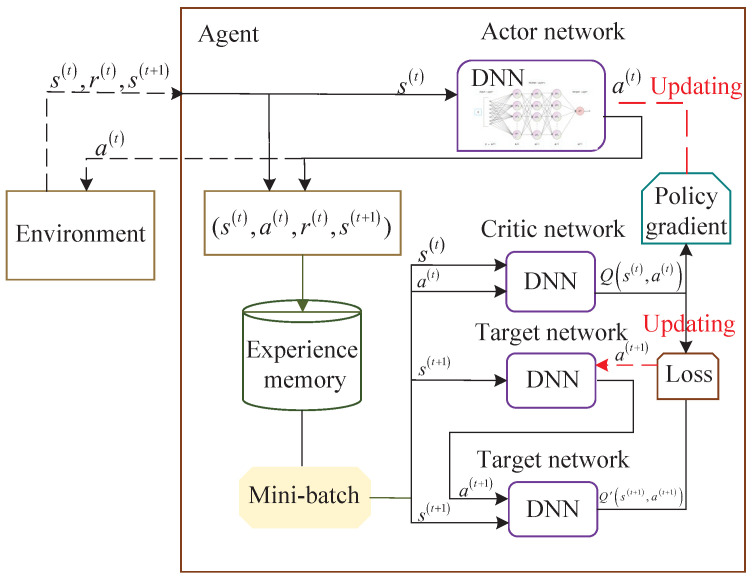
The DRL-based active transmit matrix and phase shift design framework using DDPG.

**Figure 3 sensors-23-03034-f003:**
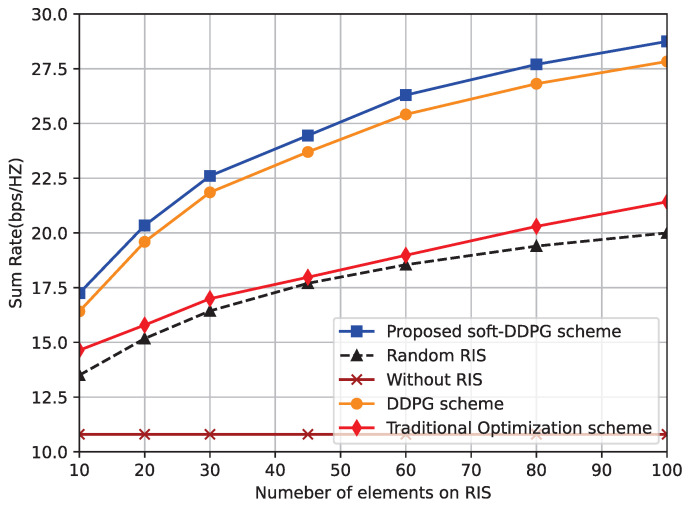
Sum rate performance relative to the increasing number of elements on RIS.

**Figure 4 sensors-23-03034-f004:**
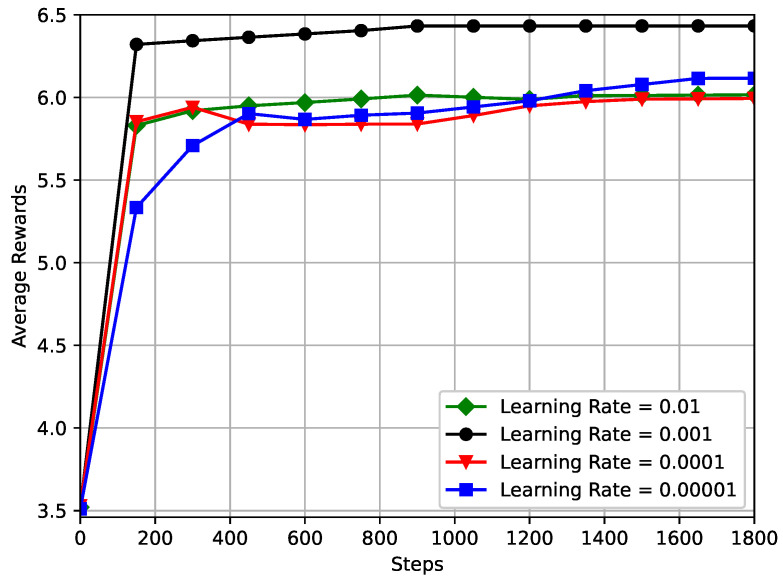
Variation of average reward under different learning rate.

**Figure 5 sensors-23-03034-f005:**
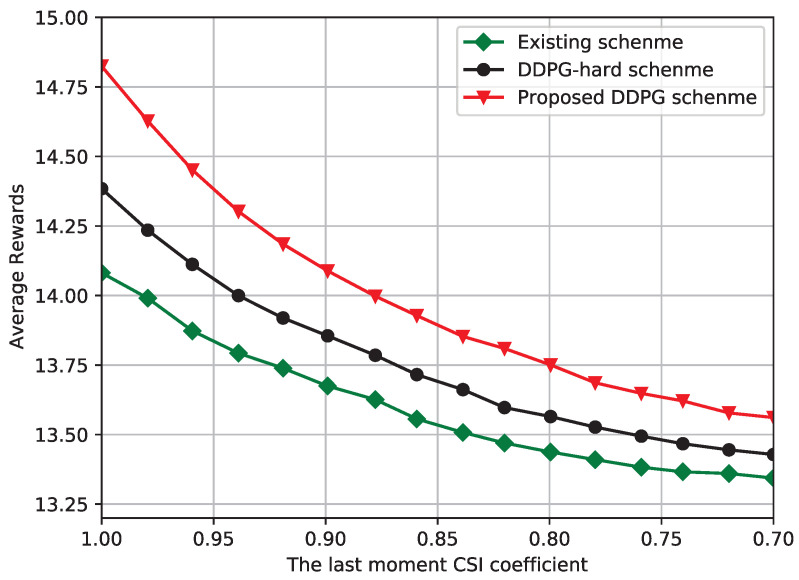
Average reward against the last moment CSI coefficient.

**Table 1 sensors-23-03034-t001:** System and DNN parameters.

System Parameters	Value
Frequency band	f=2GHz
Wavelength	λ=150mm
Noise power spectral density	−169 dBm/Hz
Link bandwidth	W=15MHz
Noise temperature	T=300 K
Height of HAP	20 km
Number of the UEs	K = 3
Transmission path	L = 3
**DNN Hyperparameters in DDPG**	**Value**
Reward discount rate	0.99
Numbers of experiences with the mini-batch	16
Learning rate	0.0001
Decaying rate	0.0001
Experience replay buffer size	100,000
Numbers of steps in each training episode	10,000

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
