# Peer review of "Joint Beamforming Design for RIS-Assisted Integrated Satellite-HAP-Terrestrial Networks Using Deep Reinforcement Learning"

_sensors, 2023, doi:10.3390/s23063034_

Round 1

Reviewer 1 Report

The work brings together an amalgam of technologies that is not argued or justified for a reason for this union. This work does not have a background that could justify and highlight the contribution. The problem formulation is confusing from my point of view and is not entirely correct. Between the ML and DL techniques, why is DRL chosen?

The channel model used is not justified, and how can they use the same one for all the space elements?

I don't see the advantage of combining so many things in one.

Reviewer 2 Report

In this work, the authors consider a RIS-Assisted integrated satellite-high altitude platform-terrestrial networks that can improve the network performance by exploiting the HAP's stability and RIS's reflection. The authors jointly optimize the transmit beamforming matrix and RIS phase shift matrix resulting in maximized system sum rate at the ground user equipment. Subsequently, the authors propose a deep reinforcement learning framework to jointly optimize the above parameters. The authors have compared the performance of the proposed method with existing methods as well.

In my opinion, the authors have addressed a good problem, however, there are few major concerns which need to be addressed before the paper can be accepted for the publication. The concerns are as follows:

1- Please provide a separate section for the related work. It helps reader to understand the background of work. Additionally, discuss some recent related work within this section. Few recent work are given below:

(a)  Joint beamforming design for ris-assisted integrated sensing and communication systems

(b)  Optimal Active Elements Selection in RIS-Assisted Edge Networks for Improved QoS

2- More result analysis need to be performed. In particular, illustrate the result analysis with various parameter variation. Discuss all the results in a detailed manner.

Overall, the authors have selected a good problem. However, a major revision is required.

Reviewer 3 Report

This paper studies beamforming design for RIS-assisted Satellite-HAP-Terrestrial Networks Using Deep Reinforcement Learning. A sum rate maximization problem is formulated, and a soft-update strategy framework based on DDPG is proposed.

I think the work is interesting. I have the following concerns.

1) The methods are not compared with traditional methods. Traditional algorithms should be provided to compare the results.

2) The writing should be improved.  E.g., HAP represents what?

3) More description of the design of state space and action space should be provided.

Round 2

Reviewer 1 Report

Many thanks to the authors for trying to address all my comments in the previous version. The work done has cleared up many previous doubts. On the other hand, I still do not see clearly the utility and advantage of using RIS with HAPs, and how these can be optimized with DRL. Perhaps, understanding the union of RIS previously with HAPS in a conventional way without DRL is better for understanding later the benefit of using DRL.

Reviewer 2 Report

The authors have addressed all my concerns. Now, the manuscript can be accepted for the publication.

Author Response

Thank you for your careful reading. 

I have checked the whole paper and corrected the typos.